# Analytical and Clinical Evaluation of Two Methods for Measuring Erythrocyte Sedimentation Rate in Eastern Indigo Snakes (*Drymarchon couperi*)

**DOI:** 10.3390/ani13030464

**Published:** 2023-01-28

**Authors:** James E. Bogan

**Affiliations:** Central Florida Zoo’s Orianne Center for Indigo Conservation, Brantley Branch Road, Eustis, FL 30931, USA; jamesb@centralfloridazoo.org

**Keywords:** erythrocyte sedimentation rate, westergren method, micro-ESR, reptile, inflammation, clinical pathology

## Abstract

**Simple Summary:**

Having a simple, reliable test to detect illness is very useful in screening animals for disease. The erythrocyte sedimentation rate (ESR) is a blood test that can detect inflammation. Although not specific for any particular disease, ESR is often used in humans as a screening “sickness indicator” due to its reliability and low cost. Little investigation of ESR in reptiles has been conducted. This study evaluates two ESR techniques in eastern indigo snakes (*Drymarchon couperi*) and found both tests performed equally. In addition, eastern indigo snakes with some inflammatory conditions had higher ESR measurements than healthy eastern indigo snakes.

**Abstract:**

Erythrocyte sedimentation rate (ESR) is a hematological test that can detect inflammatory activity within the body. Although not specific for any particular disease, ESR is often used as a screening “sickness indicator” due to its reliability and low cost. The Westergren method is a manual ESR technique commonly used but requires special graduated pipettes and over 1mL of whole blood, precluding its use in smaller patients where limited sample volumes can be obtained. A modified micro-ESR technique has been described using hematocrit capillary tubes but is used less commonly. ESR has been reported to be a useful inflammatory indicator in gopher tortoises (*Gopherus polyphemus*) and box turtles (*Terrapene* spp.) but not in Florida cottonmouth snakes (*Agkistrodon conanti*). Having an inexpensive screening test for inflammation can help guide medical decisions within conservation efforts of imperiled species. This study evaluated the correlation between these two ESR methodologies in threatened eastern indigo snakes (*Drymarchon couperi*, EIS) and found a very strong correlation (*r_s_* = 0.897), without constant or proportional biases and a reference interval of 0 (90% CI -1-1)–9 mm/h (90% CI 8-11) was defined. Additionally, a significant difference was found between healthy EIS and EIS in mid-ecdysis (*p* = 0.006) and EIS with gastric cryptosporidiosis (*p* = 0.006), indicating ESR as a useful inflammatory indicator in EIS.

## 1. Introduction

Erythrocyte sedimentation rate (ESR) is a hematological test that can detect inflammatory activity within the body caused by autoimmune diseases, infections, or neoplasia [1]. Although not specific for any particular disease, ESR is typically used in conjunction with other tests to determine increased inflammatory activity. Often, ESR can be used as a screening “sickness indicator” due to its reliability and low cost [1].

When blood is placed into a vertical column, erythrocytes (RBC) will precipitate or settle at a constant rate, which is referred to as sedimentation. During acute inflammation, inflammatory proteins cause RBCs to aggregate together and precipitate faster [1,2]. This increase sedimentation rate can be measured by a variety of manual methods or with automatic machines. The most commonly used manual method to measure ESR is the Westergren method [1]. Whole blood which has been mixed with sodium citrate is placed into a 200 mm tube with a 2.5 mm internal bore. This tube is then held in a vertical position for 1 hour at which time the degree of sedimentation is measured in millimeters. The Westergren method requires 1.25 mL of blood in order to measure ESR, limiting its use in smaller patients where that volume cannot be safely collected. As an alternative, a micro-ESR method has also been described using a hematocrit tube in a similar fashion [3].

Although not uncommonly used in human medicine, ESR is relatively underutilized in veterinary medicine and especially in reptile medicine. However, a few studies have evaluated the use of ESR in reptiles. In gopher tortoises (*Gopherus polyphemus*) and box turtles (*Terrapene* spp.), ESR has been shown to be a useful indicator of inflammation [4,5]. In Florida cottonmouth snakes (*Agkistrodon conanti*), however, ESR has not been shown to be a reliable indicator of inflammation [2].

The eastern indigo snake (*Drymarchon couperi*, EIS) is a large, diurnal colubrid native to the southeastern United States and is listed as a Threatened Species through the Endangered Species Act [6]. The EIS has some unique physiological characteristics, including extremely high plasma calcium and phosphorus levels [7,8]. Determining the normal ESR of EIS would benefit conservation efforts by allowing for monitoring of inflammatory conditions in this federally threatened species. Several inflammatory conditions have been described in both free-ranging and captive EIS including ophidiomycosis [9], pentastomiasis [10], dystocia [11], and gastric cryptosporidiosis [12]. EIS collected from the wild may have been subjected to inflammation or acute phase protein related processes and ESR may be a good screening test to determine if occult inflammatory processes, such as infectious disease, are present. Screening for infectious disease is the hallmark of good biosecurity procedures either prior to an animal entering a captive collection or prior to animal release in controlled reintroductions efforts [13,14].

The purpose of this study is to (1) compare two manual ESR methods, Westergren and micro-ESR, in EIS, (2) establish an ESR reference interval for EIS, and (3) compare the established ESR reference interval with ESR results in EIS with inflammatory conditions.

## 2. Materials and Methods

This study was approved by the Central Florida Zoo & Botanical Gardens Research Committee (Project 2022-04) and was separated into three phases. During March 2022, 17 EIS from a captive breeding colony were used in Phase 1 of this study, 12 male and 5 female. All EIS were housed according to the Association of Zoos and Aquariums Taxon Advisory Group recommended guidelines [15]. Briefly, each snake was housed individually in an 18.4 cm × 66.7 cm × 83.8 cm (7.25 in. × 26.25 in. × 33 in.) polyvinylchloride drawer and rack system (ARS, Indianapolis, IN, USA) with newsprint substrate within a dedicated room kept at 25.5 °C. A thermal gradient was not provided, and each enclosure had a polycarbonate window on one side. Lighting was available from the room’s overhead fluorescent lights and indirect sunlight through the room’s glass window which was shaded by an outside awning. The fluorescent lights were on for 8 hours a day, while the indirect sunlight allowed for a natural diurnal photoperiod. The diet was offered twice weekly and consisted of thawed frozen prey items, rotated between rats (*Rattus norvegicus*), mice (*Mus musculus*), domestic chicken chicks (*Gallus domesticus*), Japanese quail chicks (*Coturnix japonica*), and rainbow trout (*Oncorhynchus mykiss).*

All 17 EIS had been previously diagnosed with gastric cryptosporidiosis. Initial diagnosis was made with a *Cryptosporidium serpentis*-specific probe-hybridization quantitative polymerase chain reaction assay (qPCR) on a cloacal swab or gastric swab sample [16]. The diagnosis was confirmed with histological and qPCR analyses of gastric mucosal biopsies obtained with an endoscope. Blood samples were collected as part of a treatment investigational study [17].

All blood samples were collected via cardiocentesis with a 23-gauge hypodermic needle attached to a 3 mL syringe. Blood was immediately placed into a lithium heparin microtainer tube (Micro tube LH, Sarstedt AG & Co., Nümbrecht, Germany) according to manufacturer’s recommendations and thoroughly mixed by inversion. Measurement of ESR was completed within 2 hours of obtaining samples.

Two techniques were used to measure ESR, the Westergren method and the micro-ESR method. The blood samples were mixed thoroughly by inverting the microtainers 8 times prior to ESR measurements. The Westergren method used a commercial ESR kit (DISPETTEone, Guest Scientific AG, Cham, Switzerland) where 1.25 mL of whole blood was pipetted into the kit’s plastic reservoir and a 2.5 mm × 200 mm graduated measuring pipette was then inserted into the reservoir per manufacturer’s guidelines, filling the pipette by capillary action. The reservoir-pipette assembly was then placed in a rack on a level surface which ensured the tube was precisely vertical. For the micro-ESR technique, a standard non-heparinized microhematocrit capillary tube (Jorgensen Laboratories, Loveland, CO, USA) was used. The tubes were 0.9 ± 0.05 mm × 75 ± 0.05 mm. The tube was filled to the top with whole blood from the lithium heparinized microtainer then packed with 5 mm of clay, resulting in a blood volume of 0.04 mL used. The capillary tubes were placed vertically into the recesses of the hematocrit clay tray and placed on a level surface. A right angle was used to ensure the capillary tubes were precisely vertical. Both Westergren reservoir-pipette assemblies and hematocrit capillary tubes were allowed to set for 1 hour at 25.5 °C before the degree of erythrocyte sedimentation was measured in millimeters. The capillary tubes were then centrifuged (Zipocrit, LW Scientific, Lawrenceville, GA, USA) at 4400× *g* for 5 min to measure the packed cell volume (PCV).

During April and May 2018, Phase 2 of this study used 21 EIS (7 male, 14 female) that were considered healthy by physical examination and were slated to be released into the wild as part of a repatriation program. Blood samples were collected by a veterinarian during general anesthesia for radiotelemetry transponder placement and the Westergren method ESR was measured within 2 h of obtaining samples as previously described.

For Phase 3 of the study, medical records from the Central Florida Zoo & Botanical Gardens between February 2018 through November 2022 were reviewed for either Westergren or micro-ESR use in EIS. The EIS were separated into one of four groups based on the clinical diagnosis at the time of the ESR result: healthy, gastric cryptosporidiosis, ecdysis, or dystocia.

The results from the ESR measurements from all three study phases were then corrected for anemia and/or lymph dilution with Fabry’s formula [18]: ESR*_corrected_* = ESR*_measured_* × (55 − PCV*_ideal_*)/(55 − PCV*_measured_*).(1)

The mean PCV of free-ranging EIS (32%) was used as PCV*_ideal_* [7].

Analyses were computed using Excel (Microsoft 365, Redmond, WA, USA) and Real Statistics add-in software (https://www.real-statistics.com/free-download/real-statistics-resource-pack/ (accessed on 9 May 2022)). Spearman correlation coefficients (*r_s_*) were obtained to measure the linear association of Westergren and micro-ESR methods. An *r_s_* value of 0.80 to 1.0 was considered very strong correlation; 0.60 to 0.79 was moderate correlation; 0.30 to 0.59, fair correlation; 0.10 to 0.29, poor correlation; and 0.00 to 0.09, no correlation [19]. Passing–Bablok regression analysis was used to estimate constant and proportional bias between methods. Constant bias indicates that micro-ESR method consistently measures ESR to be higher or lower in comparison with the Westergren method. If the confidence interval for the y-intercept did not include the value 0, this was considered evidence of constant bias. Proportional bias indicates that the differences in measurements of each method are proportional to the level of measurement. If the 95% confidence interval for the slope did not include the value of 1, this was considered evidence of proportional bias. Bland–Altman plots were used for visualization and quantification of the agreement of the results between methods.

Distribution of ESR*_corrected_* measurements obtained from healthy EIS were evaluated for normality with kurtosis, skewness, and Shapiro–Wilk test. Outliers were determined through histogram analysis and Grubbs’ test. Results between the sexes were compared with Mann–Whitney test and if a significant difference was not found the results were combined. Guidelines for establishing reference intervals from the American Society of Veterinary Clinical Pathologists (ASVCP) were followed [20]. A reference interval was calculated using Reference Value Advisor V2.1 [21] for Microsoft Excel. The Anderson-Darling test was used to evaluate the distribution of results and non-parametric methods were used to compare medians and to define the 95% reference interval.

Results of ESR*_corrected_* measurements between EIS groups were compared. Data distribution was evaluated with kurtosis, skewness, and Shapiro–Wilk test. Comparison between the groups was performed through a Kruskal–Wallis test and post hoc analysis with paired Mann–Whitney tests. The results of normally distributed data are reported as mean, standard deviation, minimum, and maximum values; non-normal data are reported as median, minimum, and maximum values. Statistical significance was set a *p* < 0.05.

## 3. Results

The Westergren and micro-ESR techniques had good agreement as all data points on the Bland–Altman plot were within 1.96 standard deviations (Figure 1). The Spearman coefficient indicated a very strong correlation (*r_s_* = 0.897) between ESR methods. Passing–Bablok regression analysis of the methods resulted in a regression equation
(2)$$y=1.00\;\left(95\%\;\mathrm{Cl}:\;0.71-1.27\right)+0.00\;\left(95\%\;\mathrm{Cl}:-2.54\;-1.86\right)\;x$$
while the significance of linearity was acceptable (*p* = 0.799) (Figure 2). Since *p* = 0.799 > α = 0.05, it was concluded that the linearity assumption was likely to hold, without constant or proportional bias.

Two of the healthy EIS in Phase 2 were in mid-ecdysis at the time of blood collection and the results were binned in a separate cohort (mid-ecdysis). Phase 3’s medical record review revealed 16 additional cases where ESR was used in EIS: 4 healthy EIS, 6 EIS in mid-ecdysis, and 6 EIS with dystocia. In the medical record review, either Westergren method or micro-ESR methods were used. Since both methods had a very strong correlation and good agreement, all results of ESR measurements were included for comparison, regardless of the method used. A total of 54 ESR values for EIS were available for comparison (23 healthy EIS, 17 EIS with gastric cryptosporidiosis, 8 EIS in mid-ecdysis, and 6 EIS with dystocia).

The results of ESR*_corrected_* measurements in healthy EIS were normally distributed (Shapiro–Wilk method *p* = 0.233) but one significant outlier was found, and that data point was removed (Figure 3). There was not a significant difference between the healthy male and female EIS (two-tailed Mann–Whitney method *p* = 0.334) and these 22 values were combined, ranged from 0.9 mm/h to 8.7 mm/h (mean 4.5, SD 2.2), and were normally distributed (Anderson-Darling method *p* = 0.332). Due to the small sample size of 22, the data was Cox-Box transformed and robust methods were used to calculate ESR*_corrected_* reference range of 0 (90% CI -1-1)–9 mm/h (90% CI 8-11).

The results of ESR*_corrected_* measurements in EIS in mid-ecdysis cohort were not normally distributed (Shapiro–Wilk method *p* = 0.032). There was not a significant difference between male and female EIS in mid-ecdysis (two-tailed Mann–Whitney method *p* = 0.247) and these eight values were combined and ranged from 4.4 mm/h to 19.6 mm/h (median 7.1).

The results of ESR*_corrected_* measurements in EIS with dystocia cohort were not normally distributed (Shapiro–Wilk method *p* = 0.048). These six values ranged from 1.8 mm/h to 21.3 mm/h (median 5.4). 

Reference ranges for ESR*_corrected_* measurements in EIS with gastric cryptosporidiosis, EIS in mid-ecdysis, and EIS with dystocia were not calculated since the number of EIS in each group was below 20 [20].

There was a significant difference in ESR*_corrected_* measurements between EIS cohorts (Kruskal–Wallis method, *p* = 0.012). Post hoc analyses revealed there was a significant difference in ESR*_corrected_* measurements between healthy EIS, EIS with gastric cryptosporidiosis, and EIS in mid-ecdysis cohorts (paired Mann–Whitney tests *p* = 0.006 and *p* = 0.006, respectively). There was not a significant difference in ESR*_corrected_* measurements between healthy EIS and EIS with dystocia cohorts (paired Mann–Whitney test *p* = 0.614). These results are summarized in Table 1.

## 4. Discussion

This study demonstrates the micro-ESR method to have a high correlation and good agreement with the Westergren method for measuring ESR in EIS indicating the results of each method are comparable, allowing for the results of each testing method to be combined. The benefit of using the micro-ESR method over the Westergren method is the smaller volume needed to complete the test. Additionally, the capillary tube used in the micro-ESR can then be centrifuged to measure PCV minimizing the need for additional whole blood to perform a PCV as in the Westergren method. Combining the ESR and PCV measurements allows for correcting for anemia or lymph dilution, by using Fabry’s formula.

Although mammalian ESR measurements are typically completed on whole blood preserved with either ethylene diamine tetra acetic acid (EDTA) or trisodium citrate (TSC) [22], whole blood preserved with lithium heparin is described as the standard sample used for ESR measurements in reptiles [2,4,5]. When compared to TSC samples, heparinized blood will have a higher ESR measurement [23], so samples preserved with different anticoagulants should not be compared. A benefit of using heparinized whole blood over EDTA or TSC is the increased versatility of the sample. The diminutive size of many reptile patients precludes collecting large volumes of blood for multiple microtainers. For example, the smallest TSC microtainer commercially available in the US requires 0.5 mL whole blood (Blood Collection Microtube, Blue Top, Sarstedt AG & Co. KG, Nümbrecht, Germany). Whole blood treated with TSC or EDTA may be used for additional hematology, but the plasma derived from this sample should not be used for biochemical analyses as these anticoagulants can alter some biochemical analyte values in many species [24,25,26]. Furthermore, erythrocytes from some reptile species will lyse when mixed with EDTA or TSC [27]. Using only one lithium heparin microtainer allows for more versatile testing options. In addition to the micro-ESR, a complete blood cell count and plasma biochemical analytes can be performed from a single microtainer if lithium heparin is used.

A reference interval for ESR*_corrected_* was able to be calculated based on the established guidelines set forth by American Society for Veterinary Clinical Pathology [20]. Having a reference interval for ESR*_corrected_* can help the clinician determine if an inflammatory condition is present. Care must be taken, however, with interpreting the results. The presence of inflammation does not necessarily indicate a disease process as healthy EIS undergoing ecdysis may have an elevated ESR compared to healthy EIS not in ecdysis. In reptiles, ecdysis is the process of renewing the skin and this occurs over five stages where the most noticeable stages are Stage 3 and Stage 4 when the spectacles have a dull, bluish hue [28]. Starting at the end of Stage 3 and continuing through Stage 4, heterophils will infiltrate the epidermis [28]. Heterophils are also the initial cells to infiltrate reptilian tissue during inflammation [29]. Additionally, cytokines are proteins associated with the reptilian inflammatory response [30] and they are also important factors during the reptilian ecdysis cycle [31]. Therefore, it is reasonable to predict ecdysis to cause an increase in ESR values. As all of the EIS placed into the ecdysis cohort had dull, bluish spectacles, it is not surprising that the ESR was also elevated for this cohort.

Similarly, a normal ESR does not rule out a disease typically associated with inflammation. Some conditions exist that can lower ESR, such as polycythemia, sickle cell disease, and spherocytosis [1]. In addition, some conditions with known significant morbidity can have normal to low ESR values. Examples include urinary tract infection, myocardial infarction, thromboembolic disease, rheumatoid arthritis, and hypoalbuminemia [32]. The EIS cohort with dystocia did not have a significantly higher ESR when compared to the ESR of healthy EIS. This may be in line with other diseases reported to have normal ESR or it may be a product of too few cases (a type II statistical error). Further investigation into ESR and dystocia is warranted.

Additional type II statistical errors cannot be completely ruled out since the overall study size is relatively small. It is possible that some of the healthy EIS may have been experiencing some form of inflammation and falsely elevating the upper range of the ESR since the one apparently healthy EIS outlier that was removed had an ESR*_corrected_* of 12.0 mm/hr. Likewise, some EIS with dystocia may not have inflammation at the time of phlebotomy or had some comorbidity that lowers ESR, since one EIS with dystocia had ESR*_corrected_* of 1.8 mm/h while another EIS with dystocia had an ESR*_corrected_* of 21.3 mm/h.

The EIS cohort with gastric cryptosporidiosis had an elevated ESR when compared to the healthy EIS cohort. Although humans with gastrointestinal infections will often have normal ESR values [32], this elevation is not unexpected as many snakes infected with *C. serpentis* can develop severe gastric hypertrophy with marked mucosal inflammation [33]. Further studies are needed to evaluate whether ESR can play a role in determining prognosis in these cases.

Since ESR will increase with an increase in acute phase protein elevations, additional studies combining ESR and protein electrophoresis are warranted. One particular condition described in EIS, hyperviscosity syndrome, should be investigated [34]. This condition has been associated with an increase in plasma gamma globulins implicated in increasing viscosity of the blood [34] and an increase in blood viscosity can artificially decrease ESR readings in humans [35]. Further investigation is needed to determine if hyperviscosity syndrome will increase or decrease ESR in EIS.

Several factors have been identified in mammalian samples which could affect ESR results, including room temperature, time of collection, tube orientation, and vibration. An elevated room temperature may decrease blood viscosity and artificially increase the ESR [36]. Direct sunlight can also increase ESR [1]. A tilted ESR tube and increased vibrations can also increase the ESR value. An angle of three degrees from the vertical can artificially increase the ESR by up to 30% [1]. The ESR should also be performed within two hours of blood collection as a blood sample that sits too long can cause sphering and decrease the ESR value [1]. Future studies are warranted to determine if these factors affect reptile blood in the same way. In reptiles, plasma proteins can be affected by venipuncture site and season [37]. Additional studies are warranted to investigate how these variables affect ESR results in EIS.

## 5. Conclusions

Erythrocyte sedimentation rate is a simple diagnostic test that can be easily performed in-house. The micro-ESR method can be readily performed using materials already present in most veterinary practices and, in EIS, the results are comparable to the Westergren method. Adding micro-ESR to the routine CBC may allow detection of occult inflammation in EIS, especially those not undergoing ecdysis. Additionally, consideration of possible *C. serpentis* infection should be investigated in EIS with an elevated ESR.

## Figures and Tables

**Figure 1 animals-13-00464-f001:**
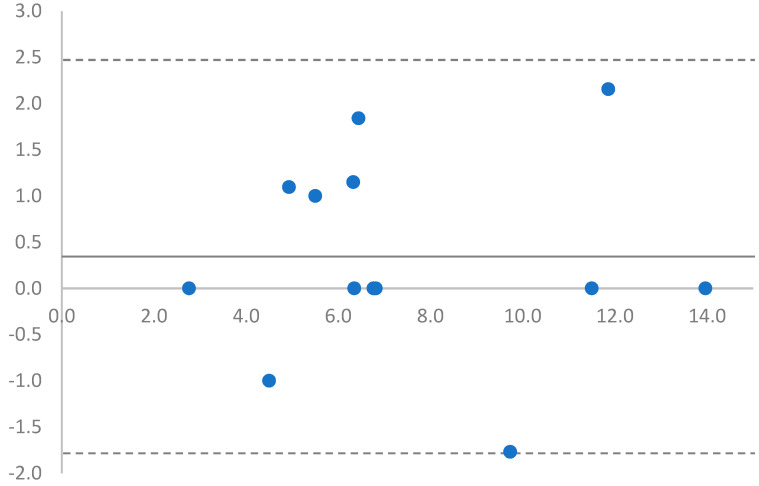
Bland–Altman Plot of erythrocyte sedimentation rate (ESR) measurements between the Westergren and micro-ESR methods. Dotted lines are ± 1.96 standard deviations from mean (solid black line). All data points are between 1.96 standard deviations signifying good agreement between methods.

**Figure 2 animals-13-00464-f002:**
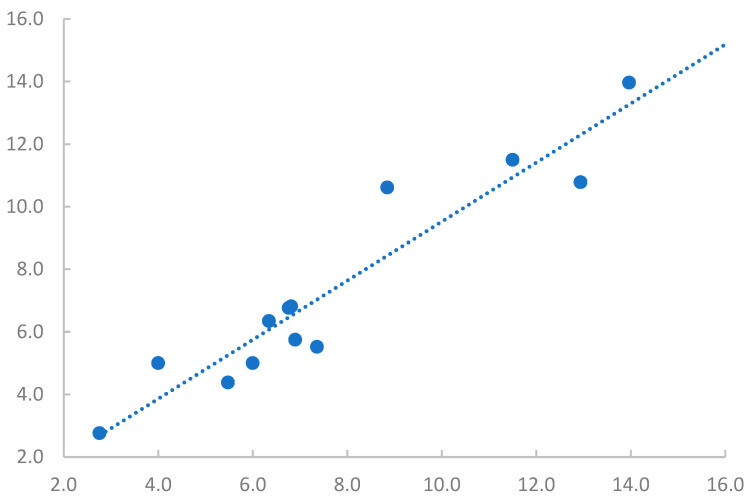
Correlation of erythrocyte sedimentation rate (ESR) measurements between the Westergren and micro-ESR methods. Spearman correlation coefficient (*r_s_*) is 0.897, signifying a very strong correlation between these two test methods. The dotted line represents the line of best fit.

**Figure 3 animals-13-00464-f003:**
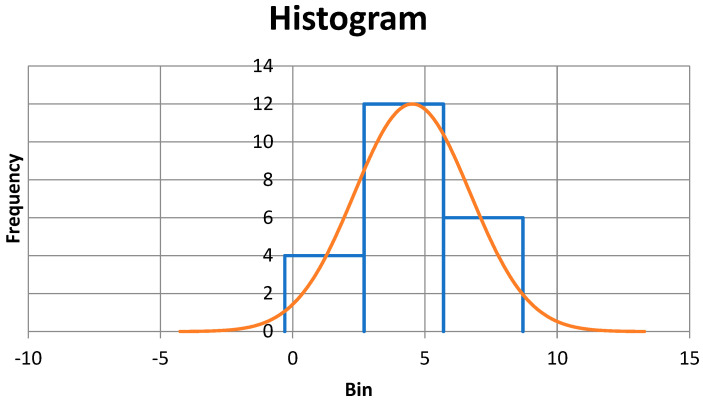
No outliers were found. There was not a significant difference between male and female EIS with gastric cryptosporidiosis (two-tailed Mann–Whitney method *p* = 0.744) and these 17 values were combined and ranged from 2.8 mm/h to 14.0 mm/h (mean 7.7, SD 3.6).

**Table 1 animals-13-00464-t001:** Comparison of corrected erythrocyte sedimentation rate (ESR*_corrected_*) measurements between healthy eastern indigo snakes (EIS) and EIS with inflammatory conditions cryptosporidiosis, ecdysis and dystocia. ESR*_corrected_* values are reported in mm/h.

EIS	n	Mean	SD	Median	Min	Max
Healthy ^1,2^	23	4.8	2.7		0.9	12.0
Cryptosporidiosis ^1^	17	7.7	3.6		2.8	14.0
Ecdysis ^2^	8			7.1	4.4	19.6
Dystocia	6			5.4	1.8	21.3

Normally distributed data are presented as mean, standard deviation (SD), minimum value obtained (Min), and maximum value obtained (Max) and non-normal data as median, Min, and Max. n = number of animals. *p* < 0.05 is considered significant and significant differences are denoted with a superscript number: 1 = ESR*_corrected_* in EIS with cryptosporidiosis was higher than in healthy EIS (*p* = 0.006), 2 = ESR*_corrected_* in EIS in ecdysis was higher than healthy EIS not in ecdysis (*p* = 0.006).

## Data Availability

The data presented in the study are available in the article.

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
