# Peer review of "Analytical and Clinical Evaluation of Two Methods for Measuring Erythrocyte Sedimentation Rate in Eastern Indigo Snakes (Drymarchon couperi)"

_animals, 2023, doi:10.3390/ani13030464_

Round 1

Reviewer 1 Report

The content of the manuscript is clinically applicable to veterinarians and animal care staff focusing on detection of health in this species. The species is also quite worthy for investigation as it has significant conservation concerns. 

My recommendations come in providing more clarity in the MM and results.  Because there are several comparisons and different groups, it is very challenging to read and understand what is being compared. 

I would recommend reconfiguration of the materials and methods to be clearer how groups were created (the MM only described healthy vs gastric cryptococcosis and indicates that a medical record search was performed) and the number of individual animals in each group. For instance I assumed that all 38 animals had both eSF methods compared, however, figure 1 only shows ~13 data points. 

I also questioned which statistical comparisons were on "pooled" micro-ESF and the standard method and which comparisons were not. 

Recommend reviewing how to show significance in Table 2, as I dont understand the currently used superscript numbers. 

Throughout text "statistically signficant" can easily be replaced with "significant" and I would remove the statistical methods used from the results, which may improve readability in this data heavy area. 

Author Response

I would recommend reconfiguration of the materials and methods to be clearer how groups were created (the MM only described healthy vs gastric cryptococcosis and indicates that a medical record search was performed) and the number of individual animals in each group. For instance I assumed that all 38 animals had both eSF methods compared, however, figure 1 only shows ~13 data points. 

Thank you for these suggestions. This section has been reformatted.

I also questioned which statistical comparisons were on "pooled" micro-ESF and the standard method and which comparisons were not. 

Thank you, this section has been edited.

Recommend reviewing how to show significance in Table 2, as I dont understand the currently used superscript numbers. 

Additional descriptions have been added in an attempt to clarify.

Throughout text "statistically signficant" can easily be replaced with "significant" and I would remove the statistical methods used from the results, which may improve readability in this data heavy area. 

Thank you for these suggestions. I have deleted “statistically” but left the methods for clarity.

Reviewer 2 Report

Dear Author,

The results presented in the Manuscript have implications for scientific and practical aspects of clinical pathology. The study design is satisfactory and easy to follow. Before the publication, additional efforts are due to resolve issues regarding the statistical methodology and the presentation of the results. Please find below the detailed comments and suggestions.

Title, abstract and keywords

- Please consider a more precise title like Analytical and clinical evaluation of the two methods for evaluating erythrocyte sedimentation rate in eastern indigo snakes (Drymarchon couperi).

- An update of the text in Lines 15‒22 might provide additional rationale for testing erythrocyte sedimentation rate (ESR) in snakes instead of the details related to the human medicine and lab methodology. The modification in the presentation of the results (Lines 23‒8) should follow the suggestions referring to the corresponding section in the text. 

- ESR is the abbreviation for erythrocyte sedimentation rate. Therefore, consider replacing ESR with another keyword. For example, the Westergren method might be a suitable choice.

Introduction

- Lines 42‒6: What were the limitations of the Westergren method necessitating the micro-ESR method development?

- Line 61: A more suitable term, like inflammation or acute phase-related processes, should replace the term host of disease processes.

- Line 67: Normal is not a necessary adjective to describe the reference interval.

Material and Methods

- The information about the observational design of the study is lacking. Also, this section should begin with an explanation of the study duration, currently included in Lines 122‒3.

- Please provide the details about the competencies of the personnel involved in the procedures with the animals.

- ESR is temperature dependent. Therefore please explain how you achieved the consistency of the ambient temperature during the measurement at the different time points.

- Lines 126‒38: The correlation analysis is not reliable for the two methods agreement assessment. Furthermore, the Pearson method, as parametric, was not suitable due to the small number of samples. Did you use results from all 38 samples in the Passing-Bablok analysis? Please cite the reference providing the interpretation of the comparison testing.

- In general, the usage of parametric statistics is questionable. The logarithmic transformation does not seem suitable due to the small number of samples. Therefore, to assure confidence in the obtained results, it would be necessary to repeat analyzes using the non-parametric methodology and modify the results and the discussion accordingly. Due to the small number of animals in total and in the subgroups, the format median with minimum and maximum would be necessary when presenting ESR results.

Results

- Please make additional efforts that this section contains the data obtained by all described methods. For example, that was not the case for the Bland-Altman analysis.

- Line 156: Please avoid quoting the correlation coefficient as a measure of the agreement between the two methods. The suggestion is also valid for the Discussion.

- Lines 159‒61: The explanation does not seem necessary.

- The description of Figure 1 differs between Line 159 and Lines 163‒5. The Figure should have marks on the axes, while the corresponding legend should explain the meaning of the dotted line.

- Lines 166‒81: The information belongs to the methodology section. Please explain why you separated snakes in mid-ecdysis from healthy ones. It does not seem reliable to present a histogram showing „normal distribution“ in a group of only 22 samples. ESR cannot have a negative value. Therefore modifications in the reference range and the corresponding confidence interval are due. Please, reconsider whether quoting the decimals had a purpose.

- Lines 185‒210: The content should be modified after the statistical retesting.

Discussion

- The first paragraph should summarize all main findings of the study.

- Lines 226‒40: Although correct, the comments seem to have limited relevance. As an improvement, consider searching for the literature data about the agreement of ESR results from the samples collected with heparin and the other anticoagulants.

References

- The number of auto citations seems above the optimal. Please consider the additional efforts to improve this aspect.

Technical Suggestions

- Line 77: Quoting the temperature in °F is not necessary.

- Information on the centrifuge Manufacturer should be part of Materials and Methods.

Author Response

The results presented in the Manuscript have implications for scientific and practical aspects of clinical pathology. The study design is satisfactory and easy to follow. Before the publication, additional efforts are due to resolve issues regarding the statistical methodology and the presentation of the results. Please find below the detailed comments and suggestions.

Title, abstract and keywords

- Please consider a more precise title like Analytical and clinical evaluation of the two methods for evaluating erythrocyte sedimentation rate in eastern indigo snakes (Drymarchon couperi).

Thank you for this suggestion. The title has been changed.

- An update of the text in Lines 15‒22 might provide additional rationale for testing erythrocyte sedimentation rate (ESR) in snakes instead of the details related to the human medicine and lab methodology. The modification in the presentation of the results (Lines 23‒8) should follow the suggestions referring to the corresponding section in the text. 

Thank you for this suggestion. This section has been modified.

- ESR is the abbreviation for erythrocyte sedimentation rate. Therefore, consider replacing ESR with another keyword. For example, the Westergren method might be a suitable choice.

Thank you for this suggestion.

Introduction

- Lines 42‒6: What were the limitations of the Westergren method necessitating the micro-ESR method development?

Thank you for this suggesstion and limitations have been added.

- Line 61: A more suitable term, like inflammation or acute phase-related processes, should replace the term host of disease processes.

Thank you for this suggestion, this has been edited.

- Line 67: Normal is not a necessary adjective to describe the reference interval.

Thank you for this suggestion, „normal“ has been deleted.

Material and Methods

- The information about the observational design of the study is lacking. Also, this section should begin with an explanation of the study duration, currently included in Lines 122‒3.

Thank you. This section has been reformatted.

- Please provide the details about the competencies of the personnel involved in the procedures with the animals.

This has been edited.

- ESR is temperature dependent. Therefore please explain how you achieved the consistency of the ambient temperature during the measurement at the different time points.

Thank you. The temperature was at the same room temperature at all time points, 25.5°C. This information has been added to the manuscript.

- Lines 126‒38: The correlation analysis is not reliable for the two methods agreement assessment. Furthermore, the Pearson method, as parametric, was not suitable due to the small number of samples. Did you use results from all 38 samples in the Passing-Bablok analysis? Please cite the reference providing the interpretation of the comparison testing.

Thank you for pointing out this error. The Pearson method has been replaced with the Spearman method. Reference citation has been added.

- In general, the usage of parametric statistics is questionable. The logarithmic transformation does not seem suitable due to the small number of samples. Therefore, to assure confidence in the obtained results, it would be necessary to repeat analyzes using the non-parametric methodology and modify the results and the discussion accordingly. Due to the small number of animals in total and in the subgroups, the format median with minimum and maximum would be necessary when presenting ESR results.

The statistical analyses have been repeated using non-parametric methods on non-logarithmically transformed data and results have been modified accordingly.

Results

- Please make additional efforts that this section contains the data obtained by all described methods. For example, that was not the case for the Bland-Altman analysis.

Done.

- Line 156: Please avoid quoting the correlation coefficient as a measure of the agreement between the two methods. The suggestion is also valid for the Discussion.

Done.

- Lines 159‒61: The explanation does not seem necessary.

- The description of Figure 1 differs between Line 159 and Lines 163‒5. The Figure should have marks on the axes, while the corresponding legend should explain the meaning of the dotted line.

Done.

- Lines 166‒81: The information belongs to the methodology section. Please explain why you separated snakes in mid-ecdysis from healthy ones. It does not seem reliable to present a histogram showing „normal distribution“ in a group of only 22 samples. ESR cannot have a negative value. Therefore modifications in the reference range and the corresponding confidence interval are due. Please, reconsider whether quoting the decimals had a purpose.

Thank you for these questions. This has been edited.

- Lines 185‒210: The content should be modified after the statistical retesting.

Done.

Discussion

- The first paragraph should summarize all main findings of the study.

- Lines 226‒40: Although correct, the comments seem to have limited relevance. As an improvement, consider searching for the literature data about the agreement of ESR results from the samples collected with heparin and the other anticoagulants.

Thank you. A statement as been added to this regard.

References

- The number of auto citations seems above the optimal. Please consider the additional efforts to improve this aspect.

Reference 8 and been replaced with a new citation (Drew, 1994). Reference 12 has been deleted. I cannot find other references to support these statements adequately.  All references have been renumbered throughout the manuscript to reflect these changes.

Technical Suggestions

- Line 77: Quoting the temperature in °F is not necessary.

This has been deleted.

- Information on the centrifuge Manufacturer should be part of Materials and Methods.

This information has been added.